# Human RNASET2: A Highly Pleiotropic and Evolutionary Conserved Tumor Suppressor Gene Involved in the Control of Ovarian Cancer Pathogenesis

**DOI:** 10.3390/ijms23169074

**Published:** 2022-08-13

**Authors:** Antonino Bruno, Douglas M. Noonan, Roberto Valli, Giovanni Porta, Roberto Taramelli, Lorenzo Mortara, Francesco Acquati

**Affiliations:** 1Immunology and General Pathology Laboratory, Department of Biotechnology and Life Sciences, University of Insubria, 21100 Varese, Italy; 2IRCCS MultiMedica, 20138 Milan, Italy; 3Department of Medicine and Surgery, University of Insubria, 21100 Varese, Italy; 4Genomic Medicine Research Center, Department of Medicine and Surgery, University of Insubria, 21100 Varese, Italy; 5Human Genetics Laboratory, Department of Biotechnology and Life Sciences, University of Insubria, 21100 Varese, Italy

**Keywords:** RNASET2, ovarian cancer, tumor suppressor genes, tumor microenvironment

## Abstract

Ovarian cancer represents one of the most malignant gynecological cancers worldwide, with an overall 5-year survival rate, being locked in the 25–30% range in the last decade. Cancer immunotherapy is currently one of the most intensively investigated and promising therapeutic strategy and as such, is expected to provide in the incoming years significant benefits for ovarian cancer treatment as well. Here, we provide a detailed survey on the highly pleiotropic oncosuppressive roles played by the human *RNASET2* gene, whose protein product has been consistently reported to establish a functional crosstalk between ovarian cancer cells and key cellular effectors of the innate immune system (the monocyte/macrophages lineage), which is in turn able to promote the recruitment to the cancer tissue of M1-polarized, antitumoral macrophages. This feature, coupled with the ability of T2 ribonucleases to negatively affect several cancer-related parameters in a cell-autonomous manner on a wide range of ovarian cancer experimental models, makes human RNASET2 a very promising candidate to develop a “multitasking” therapeutic approach for innovative future applications for ovarian cancer treatment.

## 1. Introduction

Since the proposal of the central dogma of biology by Francis Crick back in 1957 [1], ribonucleic acids have been considered for many decades as only “accessory” molecules, whose main role was to assist DNA in the work flow of genetic information from the nucleus to the cytoplasm. We now know that such a view was largely simplistic, and was based on the progressive discovery (starting from the 1990s) of a wide range of novel RNA species, endowed with an impressive range of roles in several key biological processes, including regulatory activities. This long-standing process of RNA discovery culminated in the post-genomics era, with the recognition of the central role played by RNAs in virtually every scenario in cell biology [2].

Just to emphasize the role played by RNAs in living cells, we now know that higher eukaryotes probably carry an elevated number of noncoding RNA genes compared to “classical” protein-coding genes [2]. This flood of information nowadays still represents a huge challenge for geneticists, molecular biologists and bioinformaticians, who find themselves facing a rapidly increasing world of RNA molecules showing an impressive heterogeneity in term of size, topological conformation, coding potential, subcellular localization and, most importantly, biological activities.

Ovarian cancer represents one of the most malignant gynecological tumors, with a 5-year survival rate, still ranking in the 25 to 30% range despite many years of intensive research [3,4]. Ovarian cancer is characterized by large heterogeneity, that includes epithelial ovarian cancer (EOC), germ cell ovarian cancer, stromal cell ovarian cancer and small cell carcinoma of the ovary [5,6,7]. In addition, epithelial ovarian cancer can be further classified into subtypes that include serous (low or high grade) carcinoma, clear cell carcinoma (CCC), endometrioid and mucinous cancer [8,9]. 

Currently, the main approach to epithelial ovarian cancer treatment is based on a combination of surgery and platinum-based chemotherapy. However, data from the most recent clinical trials for ovarian cancer patients suggest that conventional therapies apparently reached a plateau, since no significant increase in overall survival has been achieved in recent years [10]. Therefore, new strategies are urgently needed to overcome the limitations of currently available treatments and answering to this relevant clinical unmet need. 

The abovementioned steadily increasing interest in RNA molecules was soon followed by a concurrent attraction towards the cellular enzymatic machinery involved in RNA processing and degradation, represented by ribonucleases. Here, the number of different enzymes involved in RNA metabolism soon turned out to be impressively large as well. Within the EC2.7 subclass, where these enzymes are classified, more than 20 different types of ribonucleases have been recognized so far (either endo- or exonucleases), endowed with the ability to process single-stranded (ss) RNAs, double-stranded (ds) RNA-RNA or RNA-DNA molecules [11]. Within this frame, a growing interest has been recently addressed to ribonucleases belonging to the T2 family.

T2 RNases represent a subclass of ribonucleases endowed with a transferase-type catalytic mechanism, which generate a 2′-3′-cyclic phosphate intermediate [11] and is shared with two other classes of ribonucleases belonging to the RNase A and RNase T1 protein families, respectively [12]. Although A, T1 and T2 RNases all catalyze the cleavage of ssRNA molecules by means of the abovementioned catalytic mechanisms, several features clearly distinguish them from each other. For instance, whereas both A and T1 RNases usually show an alkaline (or weakly acidic) optimum pH for their activity, T2 RNases preferentially act at an acid pH (4–5), which might in part explain their cellular distribution. As long as the substrate specificity is concerned, class A RNases usually cut at pyrimidine sites, and class T1 enzymes appear to be more prone to guanosine cutting, whereas T2 RNases show little preference for a particular base as their cutting target site [12]. 

However, the most biologically meaningful difference between these three RNases classes probably lies in their evolutionary distribution. Indeed, whereas T1 RNase are usually restricted to bacteria and fungi and RNase A member to metazoans, T2 RNases have been found in virtually every organism, ranging from viruses to bacteria, fungi, plants, protozoans and metazoans. This extreme degree of conservation has been interpreted to suggest a key, very ancient biological role for T2 ribonucleases which, according to several reports published in the last decade focusing on T2 RNases from multiple species, seem to be related to stress response and host defense [13].

The three-dimensional structure of several members of the T2 RNase family have been reported, showing both a common α + β core structure and the presence of two highly conserved motifs (designated CASI and CASII) acting as the active site for catalysis [14]. Of note, this high structural conservation level has been observed despite both a low degree global amino acid conservation and an impressive range of biological roles.

Here, we review and discuss the cell-autonomous and non-cell autonomous role of RNASET2 in the context of ovarian cancer, as a tumor suppressor gene.

## 2. RNASET2: A Pleiotropic Tumor Suppressor Gene

The human RNASET2 gene is the only member of the T2 ribonuclease family described in our species and was cloned in 1997 [15] from chromosomal region 6q27. Of note, this region was already known to represent the site of frequent rearrangements in a wide range of human cancers [1,14,16,17,18,19,20]. In particular, a quite small region of minimal deletion was defined by Loss-of-Heterozygosity (LOH) analysis in a wide panel of human advanced ovarian carcinoma samples [21]. Noteworthy, unlike several other nearby genes, RNASET2 showed an expression pattern clearly compatible with a role in ovarian cancer pathogenesis, being expressed on the ovarian surface epithelium, inclusion cysts, and the Fallopian epithelium, which represent the structures from which most ovarian carcinomas are thought to arise. The tumor suppressive activities of RNASET2 have been reported in different tumors (Table 1). 

A TCGA survey for the RNASET2 mutation was recently carried out showing a 1 to 5% mutation rate for this gene in a wide range of human cancer types, with ovarian cancer showing as one of the highest rates [25]. According to this analysis, ovarian cancer ranks as third, in term of the alteration frequency of RNASET2. More specifically, among the RNASET2 alteration reported for ovarian cancer, 75% account as deep deletion and 25% as amplification. The higher rate deep deletion suggests a possible correlation with gene loss of function. However, no correlation with RNASET2 gene expression was reported, and the authors properly concluded that the role played by RNASET2 as a TSG might be cancer-type dependent, as widely reported for several well established oncosuppressor genes [25].

Based on these premises, the potential role of human RNASET2 as a tumor suppressor gene (TSG) involved in ovarian cancer control attracted the interest of several research groups long ago. A preliminary analysis of 55 micro dissected primary ovarian tumors samples classified from stage II to stage IV failed to reveal any clear loss-of-function mutation in the coding region of this gene [26]. However, further investigations on the same panel of primary ovarian tumors samples showed that 30% of them displayed either highly reduced or totally absent expression of RNASET2 the mRNA level. This apparent trend was even more pronounced in fourteen human ovarian cancer-derived cell lines, 86% of which showed a clear downregulation in RNASET2 expression compared to normal primary ovarian cell lines. These data suggested a role for RNASET2 as a class II TSGs, whose function is abolished in cancer tissues mainly by downregulation of its expression rather than by mutational events.

Indeed, further studies using the HEY4 human ovarian cancer cell line showed that the fraction of living clones that could be propagated in culture was constantly lower in cells transfected with an RNASET2-expression vector compared to control, empty vector-transfected clones, suggesting that the RNASET2 exerts an inhibitory effect on ovarian cancer cell growth and cloning efficiency in vitro [26]. Subsequent in vivo investigations carried out on the same cell line further confirmed the marked tumor suppressive activity of this gene, by showing its ability to suppress the tumorigenic potential of HEY4 cells in nude mice. Of note, the few tumor-producing RNASET2-transfected clones were later shown to have partially or completely lost expression of this gene, thus validating the notion of RNASET2 as a dosage-sensitive, class II TSG. Based on these preliminary data, further investigations focused on the effects of RNASET2 overexpression or silencing in several models of human ovarian carcinoma were carried out to finally unveil a highly pleiotropic oncosuppressor role carried out by this gene.

## 3. Cell-Autonomous Roles of RNASET2 as a Tumor Suppressor Gene

Early studies of the T2 ribonuclease from *Aspergillus niger* (ACTIBIND) showed the ability of this molecule to bind cytoplasmic actin and interfere with the intracellular network in human cells and to significantly decrease in vitro colony formation in several human cancer cell lines, including ovarian A2780 and breast ZR-75-1 cancer cell lines [22]. Of note, the ability of ACTIBIND to inhibit clonogenicity was independent of the enzyme’s catalytic activity. Moreover, using the ZR-75-1 cell line, the authors also evaluated the in vitro role of ACTIBIND in a tumor cancer cell invasion assay and found that cancer cell motility was also negatively affected [22]. Furthermore, ACTIBIND was also found to interfere in vitro with the process of angiogenin, as well as FGF2-induced angiogenesis by tube formation assays within vascular network in HUVEC cells [22], strongly indicating an antiangiogenic role for this member of the T2 RNase family, in addition to the previously established role as an anti clonogenic factor. Tumor angiogenesis is a well-established hallmark of cancer development, and its regulation is a key process for cancer progression and metastasis as well as for immunotherapeutics [27,28,29]. Of note, the same authors later confirmed the in vitro repressive effects on both colony formation rates and angiogenesis for human RNASET2 as well, thus suggesting a striking conservation of these oncosuppressive roles played by evolutionary distant members of the T2 RNase family [24].

The ability of human RNASET2 to negatively affect tumor cell proliferation was further confirmed by Liu et al. [30], who reported the RNASET2-induced inhibition of growth rate, cloning efficiency and anchorage-independent growth in several SV40-immortalized cell lines and a human ovarian cancer cell line. Of note, the authors also provided preliminary evidence on the involvement of the Akt pathway in *RNASET2*-mediated cancer cell growth suppression [30].

To shed light on the molecular pathways induced by *RNASET2* in human ovarian cancer cells, the gene expression profile for the Hey3Met2 ovarian cancer cell line under either high or absent *RNASET2* expression levels was later investigated [31]. This cell line was chosen due to its “asymmetrical” behavior with respect to the observed oncosuppressive role of RNASET2, which was much more evident in vivo than in vitro. Strikingly, the expression of several cancer-related genes was found to be modulated by RNASET2 even in in vitro grown cells, suggesting that this gene is able to affect several cancer pathways. Moreover, three cancer-related genes whose expression was altered in vitro by *RNASET2* modulation (*LMCD1*, *RELB* and *DSE*) showed the same pattern when assessed in vivo, thus pointing at these genes as bona fide effector genes for *RNASET2*-mediated tumor suppression.

A detailed investigation on the mode of action of *RNASET2* in ovarian cancer cells was later reported in the less aggressive, *RNASET2*-expressing OVCAR3 cell line and unveiled a highly pleiotropic role for this gene in modulating several in vitro cancer-related parameters [32]. In this work, *RNASET2* was found to act as a stress-response gene, whose expression (and secretion of its protein product in the extracellular medium) turned out to be largely increased under a wide range of stressful condition, some of them (such as metabolic and oxidative stress and hypoxia) being clearly related to the cancer microenvironment. Moreover, upon *RNASET2* knock-down (KD) by RNA interference, several in vitro parameters were clearly affected in OVCAR3 cells. For instance, *RNASET2*-KD cells showed an increased proliferation rate under CoCl_2_-induced hypoxia. Moreover, both colony formation and growth in soft agar were significantly increased in *RNASET2*-KD cells in both basal and stressful conditions, in keeping with the oncosuppressive role of this gene. Of note, another key feature of cancer cells (i.e., their ability to escape apoptosis) was impaired in *RNASET2*-KD OVCAR3 cells as well. Such response was already observed in the basal condition, but it was greatly enhanced under hypoxic stress, suggesting that the cell-autonomous oncosuppressive role of *RNASET2* is apparently exacerbated under stress conditions (such as hypoxia) that are typically experienced by early-stage cancer cells. Furthermore, in keeping with previous data reported for the *Aspergillus niger* T2 RNase (ACTIBIND), human RNASET2 was found to strongly affect the cell cytoskeleton. Indeed, whereas parental OVCAR3 cells showed a complex network of actin filaments across the cell length, such a pattern was largely disrupted in *RNASET2*-KD cells, which showed instead a strong peripheral staining of actin filament bundles, coupled with a marked rearrangement in cell shape. Of note, such drastic cytoskeletal rearrangement was rescued by adding to the culture medium recombinant RNASET2, which turned out to be effectively internalized by *RNASET2*-KD cells, thus suggesting the possible occurrence of a RNASET2-mediated paracrine effect at the tissue level regulating actin-based cell cytoskeleton changes. The functional consequences of such marked modulation of the cell cytoskeleton were also investigated and as expected, both cell migration and extracellular matrix (ECM)-adhesion pattern were found to be significantly increased in *RNASET2*-KD cells, thus including another key feature of human cancer cells (i.e., their migration and ECM adhesion rate) among those negatively regulated by the human RNASET2 protein.

Based on these data, *RNASET2*-mediated changes in gene expression profiles were investigated in OVCAR3 cells as well [33]. Of note, previous experimental data unveiled a partial localization of the RNASET2 protein in Processing-bodies (P-bodies) [34]. Since P-bodies represent intracellular sites where many types of coding and non-coding RNAs (including miRNAs) are stored, processed or degraded [35], a putative role for RNASET2 in modulating miRNA expression pattern was envisioned. Indeed, by comparing control vs. *RNASET2*-KD OVCAR3 cells, the expression pattern of several miRNAs turned out to be modulated, among which were *miR-200c* and *miR-141*, which were previously reported to be dysregulated in several human cancer types, including ovarian carcinoma [33]. The investigations on *RNASET2*-mediated gene modulation in OVCVAR3 cells was therefore extended to mRNAs expression profile [33]. Of note, almost 300 mRNAs were found to be down- or up-regulated in *RNASET2*-KD OVCAR3 cells. Most importantly, for a few downregulated miRNAs, some of their putative target mRNAs were accordingly found to be upregulated, and six of these mRNAs encoded proteins were involved in cancer-related processes. A subsequent validation by reporter gene assays using the 3′UTR regions from the genes encoding these RNASET2-modulated transcripts led to the discovery of the DDIT4L gene, a gene involved in cell growth control through the mTOR pathway, as a biological target of miR-200c, which is in turn was modulated by *RNASET2* expression levels in OVCAR3 cells. Thus, a key cancer-related feature, such as cell growth, was found to be modulated by RNASET2 in this ovarian cancer cell model.

Since most of the experimental data suggesting a role for *RNASET2* in the control of ovarian cancer growth in vitro were derived from just two cell lines, further studies were carried out in other representative experimental models. To this aim, the OAW42 and SKOV3 human ovarian cancer cell lines (characterized by high and very low endogenous *RNASET2* expression levels, respectively) were chosen for either *RNASET2* silencing by RNA interference or overexpression by stable transfection of an expression vector [36]. Of note, as already reported in OVCAR3 cells and irrespective of the applied gene modulation approach, high *RNASET2* expression levels were associated with a marked decrease in proliferation rate and anchorage-independent growth in both cell lines. Furthermore, *RNASET2* modulation in both cell lines led to a marked change in the cell cytoskeletal organization. Such rearrangements were coupled to changes in the expression of E- and N-cadherin. However, other markers of Epithelial-Mesenchymal Transition (EMT) were not significantly affected by *RNASET2* modulation in both cell lines, suggesting that the observed morphological changes were mainly attributable to actin cytoskeleton remodeling. To better investigate this topic, the growth pattern of both cell lines was also investigated in 3D cultures on different adhesion substrates, leading to the discovery that *RNASET2* expression apparently conferred to both cell lines an “ECM-dependent phenotype signature” characterized by decreased growth capability, suggesting a role for RNASET2 in modulating cell adhesion to ECM proteins and, accordingly, cytoskeletal remodeling [36]. More interestingly, the presence of such RNASET2-driven ECM-dependent signature was associated with a marked inhibition of src activation in both cell lines and, upon cell adhesion on specific ECM proteins (fibronectin for OAW42 cells and collagen I for SVOK3 cells, respectively), both AKT and MAPK phosphorylation levels increased in the absence of RNASET2 expression and were both inhibited by the src family kinase (SFK) inhibitor PP2, indicating that both signaling pathways were SFK-dependent and at the same time confirming previous studies pointing at the akt pathway in RNASET2-mediated cancer cell growth. Accordingly, assays testing the levels of phosphorylated paxillin showed the occurrence of fully mature focal adhesions colocalizing with actin only in cells with very low RNASET2 expression. Collectively, these data strongly suggested that RNASET2 expression in both ovarian cancer cell models is associated with a less aggressive tumor phenotype in terms of cell proliferation, with inhibition of ECM-dependent src kinase activation as a plausible underlying mechanism [28]. Finally, the efficacy of the PP2 src family kinase (SFK) inhibitor in these two in vitro models was also tested. Of note, RNASET2-negative cells were much more sensitive to the cytostatic effect of PP2. Altogether, these data suggested that inhibition of the SFK pathway might represent a valuable therapeutic approach for aggressive epithelial ovarian cancers.

Despite the key evidence on the oncosuppressive role of RNASET2 described above, endogenous RNASET2 expression (when evaluated at the mRNA level) in ovarian cancer cell lines and/or primary tumor was long established to vary in a wide range, from RNASET2-null to high-expressing levels. This evidence prompted Ji et al. to investigate the occurrence of RNASET2-modulating processes acting at the post-transcriptional level in ovarian cancer cells [37]. Of note, they reported the interaction of the RNASET2 protein with FBXO6 in two human ovarian cancer cell lines (A2780 and OVCAR3). FBXO6 represents one of many human F-box proteins involved in Cullin Ring Ligase (CRL)-mediated degradation of several tumor-related proteins by the ubiquitin-proteasome pathway [38]. Strikingly, both *FBOX6* knock-down and knock-out in these cell models led to a significant increase in the levels of RNASET2 protein. Conversely, *FBXO6* overexpression led to a marked decrease in RNASET2 protein expression in a dose-dependent manner, clearly suggesting that FBXO6 is involved in the degradation of RNASET2 protein in both ovarian cancer cell models. Furthermore, functional studies showed that *FBXO6* silencing in both cell lines led to a marked decrease in cell growth and colony formation. Moreover, the anti-tumorigenic effects of *FBXO6* silencing extended to the migration and invasion ability of both cell lines [37]. Of note, these functional data are totally in keeping with those previously reported in *RNASET2*-KD OVCAR3 cells in suggesting a role for RNASET2 in tumor suppression in vitro [32]. To further validate these data, the authors reported a significant upregulation of *FBXO6* in human ovarian cancer tissues. Accordingly, a subsequent survival analysis on the clinical ovarian cancer database indicated a strong correlation between higher *FBXO6* expression and the overall survival of ovarian cancer patients at advanced stages. Finally, an IHC survey on 88 human ovarian cancer specimens unveiled a negative correlation between FBXO6 and RNASET2 proteins expression, further pointing at a causal link between downregulation of RNASET2 protein in ovarian cancer and FBXO6 overexpression.

Collectively, the data reported in the previous works consistently point at *RNASET2* as a powerful oncosuppressor gene able to act upon several cancer-related features in vitro.

## 4. Non-Cell-Autonomous Roles of RNASET2 as a Tumor Suppressor Gene

Since all member of the T2 RNase family are known to encode for secreted, extracellular enzymes, a key issue to better define the biological roles of this protein class was to investigate their putative functions in the extracellular milieu. Within this frame, the role of *RNASET2* as a non-cell autonomous tumor suppressor was further investigated by in in vivo by ovarian cancer cells tumor xenografts. A first study used the *RNASET2*-null human Hey3Met2 cell line, derived from a highly metastatic subclone from the HEY4 ovarian cancer cell line [39]. This cell line was genetically engineered to overexpress the RNASET2 protein before being injected subcutaneously in nude mice.

Of note, Hey3Met2 clones overexpressing RNASET2 were clearly suppressed in their in vivo tumorigenic potential when compared to control, empty vector-transfected clones, suggesting that the marked oncosupressive role of *RNASET2* previously observed in vitro could be recapitulated in vivo as well [39]. Accordingly, a detailed histological survey of the tumors grown in vivo revealed a marked decrease in the fraction of Ki-67^+^ proliferating cells in RNASET2-overexpressing tumors, together with an increase in Cleaved-Caspase 3 (CCL3) positive cells undergoing apoptosis [39]. A massive infiltration of host-derived cells was also observed in RNASET2-overxpressing tumors, and subsequent IHC assays showed that these cells belonged to the murine host monocyte-macrophage lineage. To evaluate the functional involvement of this immune host cells population, a new Rag2^−/−^γ_c_^−/−^ xenograft-based model was assembled whereby the same Hey3Met2 clones were inoculated in vivo. As expected, *RNASET2*-overexpressing tumors were still significantly suppressed in their growth rate. Moreover, pretreatment of the mice with a macrophage-depleting agent (clodronate liposomes) significantly impaired the ability of RNASET2 to suppress tumor growth in vivo, thus suggesting that host macrophages played a key role in *RNASET2*-mediated tumor suppression [37].

Tissue macrophages have long been established as key immune effector cells in the tumor microenvironment (TME), where they are mostly associated with pro-tumor activities, ranging from immunosuppression to angiogenesis [28,29,40,41,42,43], especially in advanced cancers [42]. On the other hand, this subpopulation of human innate immune cells is also known to be endowed with an impressive functional plasticity, which in turn allows them to carry out either pro- or anti-tumoral activities in vivo, mainly depending on both cancer cells- and TME-derived cues [44]. Interestingly, by investigating the polarization state of tumor-infiltrating macrophages in *RNASET2*-expressing tumors, most of the infiltrating macrophages were found to belong to the M1 class [29], which has long been known to play a strong antitumor role [30] role. These data clearly support the notion of a completely novel and independent oncosuppressive role played by human RNASET2 in vivo, which appeared to be different from those previously observed in in vitro experimental models as described above. According to this new model, the recruitment of innate immune cells, belonging to the macrophage lineage, could be envisaged as a key process in *RNASET2*-mediated tumor suppression in vivo.

However, some notes of caution on the above-mentioned in vivo data stemmed from the use of a poorly known ovarian cancer cell line with an RNASET2-overexpression experimental plan, which might have biased the observed data by establishing an unphysiological setting. Therefore, the in vivo oncosuppressive role of *RNASET2* was further investigated in a more conventional ovarian cancer cell model, represented by the OVCAR3 cell line. Of note, the endogenous high levels of RNASET2 expression in these cells allowed the use of an alternative genetic manipulation approach, based on gene knockdown by RNA interference rather than unphysiological gene overexpression [45].

*RNASET2*-silenced OVCAR3 clones were tested in nude mice together with control, scrambled siRNA-transfected clones. Noteworthy, *RNASET2*-silenced clones developed into fast-growing tumors, with a marked increase in growth rate compared to control clones, in keeping with the in vivo oncosuppressive role previously reported in Hey3Met2 cells [45]. A detailed histological analysis of tumor samples showed a marked decrease of host-derived cells in RNASET2-silenced tumors, and IHC assays confirmed that the depleted population of cells mainly consisted of M1-polarized macrophages [45]. Thus, the key role of macrophages in in vivo *RNASET2*-mediated tumor suppression was confirmed in an independent ovarian cancer experimental model. To shed further light on the molecular pathways involved in tumor suppression by human *RNASET2*, the gene expression profile of both human cancer cells and mouse stromal cells in control vs. RNASET2-silenced tumors was evaluated by means of microarray hybridization [45]. In keeping with the observed in vivo effects, human genes regulated by RNASET2 turned out to be significantly enriched in functional classed related to leukocyte activation, although several genes involved in development were also highly represented. Moreover, most genes involved in leukocyte activation were found to be downregulated in *RNASET2*-silenced tumors, suggesting a positive regulatory function for RNASET2 on the immune and inflammatory system [45]. As long as the mouse host gene expression profile was concerned, two pathways clearly emerged when comparing *RNASET2*-silenced and control tumors: cell adhesion and immune response [45]. Whereas the first pathway is in keeping with previously described results in in vitro experimental settings, the second one was again pointing at an independent, noncell-autonomous oncosuppressive role for RNASET2 related to immune response activation, being particularly enriched in functional categories such as “innate immune response”, “inflammatory response” and “host defense”. Moreover, subsequent in vitro assays using a recombinant RNASET2 protein unveiled its ability to act as a powerful chemotactic agent for peripheral blood leukocytes (PBL)-derived human macrophages, thus providing a further validation of the role played by macrophages in *RNASET2*-mediated tumor suppression in vivo [45]. Of note, the absolute requirement for RNASET2 secretion by cancer cells to induce tumor suppression in vivo had been previously demonstrated by showing that, by adding a KDEL endoplasmic reticulum retention signal to the RNASET2 protein, its ability to trigger tumor suppression in nude mice was completely suppressed [32].

These data deals with the identification of RNASET2 alarmin molecule, according with the proposed model that cells from early-stage neoplastic lesions undergoing cancer-related stressful conditions (such as hypoxia or nutrient starvation), actively increase RNASET2 protein expression and secretion in the TME, where it acts as a M1-polarizing agent for resident tissue macrophages and/or recruited circulating monocytes. Furthermore, the T2 RNase from the trematode parasite Schistosoma *mansonii* (omega-1) has been reported to prime host dendritic cells (DCs) to trigger a Th2 response [46], following internalization by DCs through the mannose receptor [46]. Interestingly, mannose receptor represents a surface marker typically expressed by M2 macrophages, suggesting that RNASET2-mediated macrophage polarization might be induced by mannose receptor-mediated uptake by these cells, followed by a reprogramming toward a M1-polarization pattern.

Taken together, these results suggested that the RNASET2 protein is able to establish a crosstalk between cancer cells and cellular effectors of the TME in order to raise a potent anticancer innate immune response. On this perspective, RNASET2 was shown to apparently act in vivo as an alarmin-like molecule [25,47], leading the authors to speculate that, under early cancer cell-induced stressful condition experienced by the TME (such as hypoxia or nutrient starvation), the resulting increase in RNASET2 expression and secretion by cancer cells triggers a host defense response involving the recruitment and functional activation of M1 macrophages endowed with antitumoral activity. In turn, these data added further evidence to the notion that host macrophages may play different and even opposite roles in cancer biology. Indeed, tumor-associated macrophages (TAMs) have long been known to represent a population of cancer-promoting immune cells in several tumor types (including breast and ovarian cancers), being involved in immune suppression, angiogenesis, metastasis [5,40,41] and regulation of angiogenesis by immune cells [48]. On the other hand, macrophages are also equipped with antitumor features, enabling them to kill tumor cells and modulate the acquired immune system accordingly. These opposing roles reflect the high intrinsic plasticity of tissue macrophages that are able to acquire different phenotypes and effector roles in response to specific microenvironmental cues, resulting in a continuum of distinct polarization states whose extreme states are represented by the M1-like and M2-like subtypes, endowed with anti-tumor and pro-tumor activity, respectively [28,29,40,41,42,43,44,47]. Since TAMs usually present M2-like features with heterogeneous phenotypes and functions, the fine tuning of the M1/M2 balance has long been considered as a key target for cancer immunotherapy [28,29,40,41,42,43,44,47]. In this context, it is worth noting that RNASET2 is able to influence the M1/M2 balance of macrophage polarization in vitro. Indeed, when the monocyte-like THP-1 human cell line (which expresses high levels of endogenous *RNASET2*) was differentiated into M0 macrophages and subsequently polarized with appropriate stimuli, the M1 polarization response was much more pronounced compared to the M2 alternative state [49]. However, following *RNASET2* knockdown by RNA interference, THP-1-derived macrophages were strongly inhibited in their response to M1-polarizing factors and at the same time showed a marked M2 polarization profile when induced with M2 stimuli. Of note, treatment of *RNASET2*-silenced THP-1 cells with recombinant RNASET2 showed a partial but consistent rescue to the parental M1 macrophage polarization pattern, whereas M2-polarized cells displayed a weakened response, further suggesting that tissue macrophages are sensitive not only to their endogenous level of RNASET2 expression [49], but also to exogenously provided RNASET2, in keeping with the alarmin hypothesis.

Noteworthy, a murine Rnaset2 syngeneic murine model was also recently reported by using mouse *Rnaset2*-overexpressing C51 colon carcinoma and TS/A mammary adenocarcinoma syngeneic cells in comparison to control empty vector-transfected tumor cells [50]. Although the reported in vivo study was mainly focused on the C51 cancer cell model, a trend for a Rnaset2-mediated tumor suppressive effect was observed in TS/A cells as well [50]. Of note, in the in vivo C51-based syngeneic model, a strong *Rnaset2*-dependent retardation in tumor growth rate was observed, concomitant with early tumor infiltration by M1 macrophages, inhibition of M2 and myeloid-derived suppressor cells (MDSCs) cells and, very interestingly, a later expansion of CD8^+^ T lymphocytes with rejection capacity and antitumor recall immunity [50]. Further experiments will be required in the hormone-sensitive TS/A-based breast syngeneic mouse cancer model, in which both CD4^+^ and CD8^+^ T cells seems to have a major antitumor role [51,52].

## 5. Concluding Remarks

The last two decades witnessed a more and more increased interest in the biology of *RNASET2* in different cancer types, including ovarian cancer. Much evidence, coming from different experimental models and tumor types, now also allow for proposing RNASET2 as a very promising candidate molecule for ovarian cancer characterization and potential therapy. In particular, the consistent findings of the highly pleiotropic oncosuppressive role of RNASET2 in ovarian cancer (acting at both cell-autonomous and noncell-autonomous levels and thereby targeting both cancer cells and a key component of the TME innate immune system such as tissue macrophages) hold great potential for the introduction of this multi-faceted oncosuppressor protein in the setting of innovative molecules endowed with the ability to hamper cancer cell growth on multiple levels of action. In particular, the demonstrated role of this ribonuclease in modulating macrophage’s polarization pattern in vivo might pave the way for a novel RNASET2-based immunotherapy approach, placing RNASET2 as an immunoregulatory molecule endowed with alarmin-like functions. Indeed, the development of therapeutic strategies aimed at re-educating M2 macrophages towards the M1 counterpart, within the TME, has long been established as a promising therapeutic approach for cancer treatment [43]. This perfectly fits the concept of immunotherapy, referred as the possibility to re-activate the host immune system, that has been blocked by tumor cell and tumor cell interactions within the TME.

The overall interest for the tumor microenvironment regulation of cancer induction and progression, together with the recognized role of the TME as a target for therapy culminated with the advent of immunotherapy, are potential breakthrough therapeutical approaches. The advent of immunotherapy has also been promoted by significant advances in our knowledge on the cellular and molecular bases of immune regulation of cancer cells growth. Accordingly, several immune checkpoint inhibitors have been recently approved for a wide range of cancers [53,54,55,56]; however, as long as ovarian cancer is concerned, immunotherapy is still in its “pre-birth” and there are currently no data from clinical trials based on this approach. In this context, RNASET2 might therefore contribute to a still underrepresented area of research and represent a novel and innovative tool for the development of a novel immune system-based treatment for this cancer type.

Of note, besides the remarkable amount of data gathered on experimental cancer models described above, several lines of evidence further support the potential of *RNASET2* as a promising tool for ovarian cancer therapy. For instance, a marked correlation between *RNASET2* expression and overall survival (OS) in epithelial ovarian cancer patients has been reported in two independent datasets, whereby patients whose tumors displayed low RNASET2 expression showed a significantly shorter OS compared to high-RNASET2 expression patients [50].

In keeping with the above-mentioned data, by comparing gene expression in carboplatin-resistant (A2780-CBP) and cisplatin-resistant (A2780-DDP) ovarian cancer cells and their drug-sensitive counterparts (A2780), Yin et al. found that *RNASET2* expression was significantly decreased in drug-resistant cells [57], thus suggesting a role for RNASET2 in drug resistance. Further in vivo data from human ovarian cancer tissues similarly showed a decreased RNASET2 expression in drug-resistant versus drug-sensitive tissues [57]. Moreover, using GeneMANIA web-based database software, the authors were able to predict gene/protein-gene/protein interactions and found a direct interactions of RNASET2 with several genes or proteins involved in drug resistance in ovarian cancer, such as checkpoint kinase 2 (CHEK2), programmed cell death 4 (PDCD4), phosphatase and tensin homolog (PTEN), split hand/foot malformation (ectrodactyly) type 1 (SHFM1) and Yes-associated protein 1 (YAP1) [57]. Of note, a potential mechanism of drug-resistance mediated by altered RNASET2 expression could be related to the activation of the PI3K/Akt signaling pathway, as previously suggested by other authors.

In conclusion, the above-mentioned experimental evidence converges on the notion of RNASET2 as a multi-faceted, highly pleiotropic oncosuppressor protein in the context of human ovarian cancer. We therefore reckon that future studies, particularly on human clinical samples, should implement the potential consideration of RNASET2-based therapy for the treatment of one of the most life-threatening cancers striking millions of women worldwide.

## Figures and Tables

**Table 1 ijms-23-09074-t001:** Published articles reporting a tumor suppressive role for T2 RNases in human cancer types.

T2 RNase Family Member	Human Cancer Type	References
ACTIBIND (*A. niger*)	Colon, breast, ovary, angiogenesis	[22]
ACTIBIND (*A. niger*)	Melanoma	[23]
RNASET2 (h. sapiens)	Colon cancer	[24]
RNASET2 (h. sapiens)	Prostate cancer	[1]

## Data Availability

Not applicable.

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
