# Peer review of "Human RNASET2: A Highly Pleiotropic and Evolutionary Conserved Tumor Suppressor Gene Involved in the Control of Ovarian Cancer Pathogenesis"

_ijms, 2022, doi:10.3390/ijms23169074_

Round 1

Reviewer 1 Report

The work is of moderate interest as it lists data, most of these functional from cell line models, mainly produced by the (same) authors.

The evidence for RNASET2 as TSG in OVCA is not overwhelming.

There are no genomic data really supporting for the TSG role. For loss of expression there is no explanation -aberrant methylation, oncogenic regulation. There is no animal model of spontaneous or induced tumorigenesis. So in general little interest.

To increase the interest it would be great if the authors could re-analyse (metanalyse) genomic and gene expression datasets such as TCGA of OVCA to search for somatic mutations and correlations of lost expression.

Is there more evidence for TSG role in othere cancers?

Although this is primarily a re view it should somehow defend the significance of RNASET2. 

Author Response

The work is of moderate interest as it lists data, most of these functional from cell line models, mainly produced by the (same) authors.

Response: The aim of our work was exactly to provide the state of the art for RNASET2 biology in ovarian cancer. Therefore, a detailed analysis and discussion for the functional roles was necessary, which includes in vitro, in vivo studies and those few available in humans. RNASET2 is a molecule largely studies by the authors of the present work, that provided relevant insights in the specific field. Therefore, is not unexpected that many publications cited are by the same authors.

The evidence for RNASET2 as TSG in OVCA is not overwhelming.

Response: This is a crucial point, suggesting that further studies are with no doubt required to dissect the role of RNASET2 in ovarian cancer, based on both in vitro and in vivo evidence. However, ovarian cancer is the human cancer type with the higher number of reports (by different research groups) strongly supporting the notion of RNASET2 as a tumor suppressor gene.

There are no genomic data really supporting for the TSG role. For loss of expression there is no explanation -aberrant methylation, oncogenic regulation. There is no animal model of spontaneous or induced tumorigenesis. So, in general little interest.

Response: Although the molecular mechanisms underlying RNASET2 downregulation in human cancers have not been established yet, we reckon that the bulk of in vitro and in vivo experimental data gathered on the pleiotropic oncosuppressive functions of T2 RNases in ovarian cancer (see the detailed references in the text supporting the statements), coupled to ovarian cancer patients’ survival rates analysis, strongly indicate a key role for this gene in ovarian cancer suppression.

Rat and mouse gene knockout models have been reported only very recently (doi:10.1242/dmm.032631; doi: 10.1038/s41467-021-26880-x) due to a gene duplication of the Rnaset2 gene in the rodent lineage. Although in both models the authors did not assess the frequency of spontaneous cancer occurrence it is worth noting that, due to the well-established role of RNASET2 as a stress-response gene, an experimental model more suitable to validate the role of this gene as a TSG in vivo would require an induced tumorigenesis approach, which to our knowledge has not been attempted so far.

To increase the interest, it would be great if the authors could re-analyse (metanalyse) genomic and gene expression datasets such as TCGA of OVCA to search for somatic mutations and correlations of lost expression.

Response: A TCGA survey for RNASET2 mutation was recently carried out (doi: 10:3389/fimmu.2020.01554) who reported a 1 to 5% mutation rate for this gene in a wide range of human cancer types, with ovarian cancer showing one of the highest rates. However, no correlation with RNASET2 gene expression was reported, and the authors properly concluded that the role played by RNASET2 as a TSG might be cancer-type dependent, as widely reported for several well established oncosupressor genes.

Is there more evidence for TSG role in other cancers?

Response: A table showing published articles reporting a tumor suppressive role for T2 RNases in human cancer types different from ovarian carcinoma, has been integrated in the text (Table 1).

Although this is primarily a review, it should somehow defend the significance of RNASET2. 

We are confident that RNASET2 biology in cancers is still a challenge. As for all the molecules to be proposed as key regulator of cancer and/or possible biomarkers, there still lot of research to be conducted. While stressing on the literature currently present in the specific topic, we admitted some limitation that have been indicated in the text.

Reviewer 2 Report

In this review, authors summarized the role of RNASER2 in pathogenesis of ovarian cancer with emphasis on recruitment and polarization of macrophages. Generally, this review properly summarizes the background of RNASET2 and the related studies, and clearly addresses the anticancer role of RNASET2. Several suggestions are listed for the improvement of the present manuscript.

1. “RNAse” is suggested to be corrected as “RNase”

2. The reference(s) for FBXO-mediated RNASET2 degradation in ovarian cancer is missing.

3. How does RNASET2 of ovarian or other cancer cells regulate macrophage polarization is an interesting issue and should be further discussed.

4. Too many “Strikingly”, authors may further improve some wording in the manuscript.

Author Response

In this review, authors summarized the role of RNASER2 in pathogenesis of ovarian cancer with emphasis on recruitment and polarization of macrophages. Generally, this review properly summarizes the background of RNASET2 and the related studies, and clearly addresses the anticancer role of RNASET2. Several suggestions are listed for the improvement of the present manuscript.

  1. “RNAse” is suggested to be corrected as “RNase”

Response: Revised

  1. The reference(s) for FBXO-mediated RNASET2 degradation in ovarian cancer is missing.

Response: The required reference has been integrated in the text.

  1. How does RNASET2 of ovarian or other cancer cells regulate macrophage polarization is an interesting issue and should be further discussed.

Response: The role played by RNASET2 in the modulation of human macrophages polarization in the context of human ovarian cancer has been thoroughly described in the manuscript. According to the alarmin model proposed by the authors, cells from early-stage neoplastic lesions undergoing cancer-related stressful conditions (such as hypoxia or nutrient starvation) actively increase RNASET2 protein expression and secretion in the TME, where it acts as a powerful M1-polarizing agent for resident tissue macrophages and/or blood monocytes. Noteworthy, the T2 RNase from the trematode parasite Schistosoma mansonii (omega-1) has been reported to prime host dendritic cells (DCs) to trigger a Th2 response (doi: 10.1084/jem.20082460, now integrated in revised text), following internalization by DCs through the mannose receptor (doi: 10.1084/jem.20111381, now integrated in revised text). Interestingly, mannose receptor represents a surface marker typically expressed by M2 macrophages, suggesting that RNASET2-mediated macrophage polarization might be induced by mannose receptor-mediated uptake by these cells, followed by a reprogramming toward a M1-polarization pattern.

  1. Too many “Strikingly”, authors may further improve some wording in the manuscript.

Response: The text was revised and implemented, thanks to the participation of all the authors involved. In the text drafting.

Round 2

Reviewer 1 Report

I would like the authors to provide full description of these data: "A TCGA survey for RNASET2 mutation was recently carried out (doi: 10:3389/fimmu.2020.01554) who reported a 1 to 5% mutation rate for this gene in a wide range of human cancer types, with ovarian cancer showing one of the highest rates" So what variants and of what consequence were found? They do not have to correlare with expression, but thet can still be loss-of-function.

Author Response

Reviewer's comment. I would like the authors to provide full description of these data: "A TCGA survey for RNASET2 mutation was recently carried out (doi: 10:3389/fimmu.2020.01554) who reported a 1 to 5% mutation rate for this gene in a wide range of human cancer types, with ovarian cancer showing one of the highest rates" So what variants and of what consequence were found? They do not have to correlare with expression, but thet can still be loss-of-function.

Response by authors: we thank the reviewer for the comment.  According to the analisis in  10:3389/fimmu.2020.01554, ovarian cancer ranks as third, in term of the alteration frequency of RNASET2. More specifically, among the RNASET2 alterations reported for ovarian cancer,  over 75% account as deep deletions and over 25% as amplifications. The higher rate deep deletions suggests a possible correlation with gene loss of function.